# Characterization of intestinal *O*-glycome in reactive oxygen species deficiency

Radka Saldova[1,2,3]*, Kristina A. Thomsson[4], Hayden Wilkinson[1,2,3], Maitrayee Chatterjee[3¤], Ashish K. Singh[3], Niclas G. Karlsson[5,6], Ulla G. Knaus[3]

**1** National Institute for Bioprocessing, NIBRT GlycoScience Group, Research and Training, Blackrock, Dublin, Ireland, **2** CÚRAM, SFI Research Centre for Medical Devices, National University of Ireland, Galway, Ireland, **3** School of Medicine, University College Dublin, Dublin, Ireland, **4** Proteomics Core Facility, Sahlgrenska Academy, University of Gothenburg, Gothenburg, Sweden, **5** Department of Medical Biochemistry and Cell Biology, Institute of Biomedicine, Sahlgrenska Academy, University of Gothenburg, Gothenburg, Sweden, **6** Faculty of Health Science, Department of Life Science and Health, Oslo Metropolitan University, Oslo, Norway

¤ Current address: The TIM Company, Delft, Netherlands
* radka.fahey@nibrt.ie

## Abstract

Inflammatory bowel disease (IBD) is characterized by chronic intestinal inflammation resulting from an inappropriate inflammatory response to intestinal microbes in a genetically susceptible host. Reactive oxygen species (ROS) generated by NADPH oxidases (NOX) provide antimicrobial defense, redox signaling and gut barrier maintenance. NADPH oxidase mutations have been identified in IBD patients, and mucus layer disruption, a critical aspect in IBD pathogenesis, was connected to NOX inactivation. To gain insight into ROS-dependent modification of epithelial glycosylation the colonic and ileal mucin *O*-glycome of mice with genetic NOX inactivation (*Cyba* mutant) was analyzed. *O*-glycans were released from purified murine mucins and analyzed by hydrophilic interaction ultra-performance liquid chromatography in combination with exoglycosidase digestion and mass spectrometry. We identified five novel glycans in ileum and found minor changes in *O*-glycans in the colon and ileum of *Cyba* mutant mice. Changes included an increase in glycans with terminal HexNAc and in core 2 glycans with Fuc-Gal- on C3 branch, and a decrease in core 3 glycans in the colon, while the ileum showed increased sialylation and a decrease in sulfated glycans. Our data suggest that NADPH oxidase activity alters the intestinal mucin *O*-glycans that may contribute to intestinal dysbiosis and chronic inflammation.

## Introduction

Inflammatory bowel disease (IBD), a chronic inflammatory disorder with rising prevalence worldwide, includes mainly Crohn's disease and ulcerative colitis [1]. IBD is a multifactorial disease that may result from an inappropriate inflammatory response to intestinal microbes in a genetically susceptible host, with altered host-microbe interactions likely driving pathogenesis [2]. Intimately involved in reciprocal signaling between microbe communities and host are epithelial NADPH oxidases (NOX/DUOX) that provide not only maintenance and restitution

**Data Availability Statement:** Source data are available upon request. Raw MS data for structural interpretation of reduced nonderivatized O-glycans

from ileum and colon using PGC-MS are available on GlycoPOST (https://glycopost.glycosmos.org/) using the Project ID no. GPST000307 and GPST000386 and MSMS data with tentative structures are available https://unicarb-dr.glycosmos.org/references/521.

**Funding:** This work was supported by Science Foundation Ireland (SFI, https://www.sfi.ie/) co-funded under the European Regional Development Fund under grant number 13/RC/2073 (RS, HW), Science Foundation Ireland 16/IA/4501 (UGK).

**Competing interests:** The authors have declared that no competing interests exist.

of the gut barrier, but also protection against pathogens. Several NADPH oxidases are expressed in colonocytes, intestinal stem cells and secretory cells where they maintain various biological functions by generating superoxide or hydrogen peroxide for redox signaling [3]. Loss-of-function variants of NOX1, NOX2 and DUOX2 have been linked to pediatric IBD and CGD-IBD, respectively [4–6]. Mice with single Nox1, Nox2 (*Cybb*) or Duox2 deletion or inactivation do not reflect human IBD pathogenesis, likely due to the multifactorial nature of colitis that cannot be mimicked in the protective standards of animal facilities [3]. Susceptibility to barrier disruption and epithelial inflammation was achieved when providing a second hit (2-hit model), for example by crossing Nox2 and Nod2 knockout mice [7] or by establishing reactive oxygen species (ROS) deficiency in both the epithelial and the immune compartment by introducing a missense mutation in the NOX dimerization partner protein p22$^{phox}$ (*Cyba*) [8]. This *Cyba* mutation inactivates three Nox isoforms (Nox1-3), resulting in substantially decreased ROS generation, and mimics a mutational hot spot in chronic granulomatous disease (CGD) patients. *Cyba$^{nmf333}$* mice (*Cyba* mutant mice) were highly susceptible to severe colonic inflammation when challenged due to the underlying mucus, microbiota, and innate immune defects [8].

Glycosylation is the most common posttranslational modification of proteins. Changes in glycans were found in many diseases including chronic inflammatory diseases and cancer [9]. Mucins are glycosylated proteins that constitute the main components of mucus on the surface of mucosal epithelial cells [10]. In the intestine the predominant mucin is MUC2 which is *N*- and *O*-glycosylated, with *O*-glycosylation being far more abundant. Glycosylation permits water molecule incorporation for gel formation and restricts access of mammalian and bacterial proteases to the mucin protein core [11]. Mucins have multiple roles within the intestinal mucosal barrier as they form a dense protective layer in the colon to prevent the entry of pathogenic organisms/substances, but also provide in the loose mucus layer glycosylated attachment sites for commensal bacteria, enabling colonization with a mutually beneficial mucus-associated microbiota [12]. By maintaining the impenetrable dense mucus layer, mucin-type *O*-glycans protect from spontaneous colitis in mice [13]. Intestinal inflammation is associated with changes in genes regulating epithelial glycosylation and alterations in synthesis, packing, storage and release of MUC2 mucin. Mucin analysis in IBD patients and mice challenged with colitis inducers revealed a simplified mucin *O*-glycome with truncated *O*-glycans and changes in terminal fucosylation, sialylation and sulfation [12,14,15]. These alterations contribute to mucin degradation, increased barrier permeability, microbiota modification and inflammation [12,14]. Fucosyltransferases (*FUT*) catalyze the transfer of L-fucose to various sugar acceptor substrates, and the resulting fucosylation plays an important role in signaling pathways, mucin modification and mucus-bacteria interactions. Triple α1,2 fucosyltransferase knockout mice (*Fut1$^{-/-}$Fut2$^{-/-}$Sec1*(Fut3)$^{-/-}$ mice) are completely deficient in H antigen on red blood cells. While mucin *Muc2* and *Fut1* expression were not altered in *Cyba* mutant mice, the expression of *Fut2* was decreased by 30–40% [8]. The α1,2-fucosyltransferase FUT2 catalyses the addition of fucose to terminal galactose, and in humans *FUT2* nucleotide polymorphisms are associated with changes in the intestinal microbiota composition and predisposition to Crohn's disease [16]. Genetic variation in *FUT2* determines the ABH secretor status and in cooperation with the α-(1,3/4)-fucosyltransferase FUT3 the Lewis status. *FUT3* variants are also considered an IBD risk factor, likely due to dysbiosis as fucosylated glycans serve as attachment sites and carbon source for microbiota.

To gain insight in how ROS deficiency may weaken the mucus barrier by altering the *O*-glycome, we characterized and compared the colonic and ileal mucin *O*-glycans in *Cyba* mutant mice with internal control mice.

## Materials and methods

### Mice

B6 *Tyr*⁺- *Cyba*^*nmf333*/J mice (JAX#005445, [17]) were backcrossed to in-house bred C57BL/6J mice for more than 10 generations. Internal control *Cyba*^+/+ (WT) mice were prepared by separating genotypes at the F2 generation, derived from F1 heterozygotes, and using separately maintained F3 and F4 generations for this study. All mice were housed in individually ventilated cages in the same room in an SPF facility operating at FELASA standards and were supplied with Teklad 2018 diet (Envigo) and sterile water *ad libitum*. All animal experiments were performed in accordance with EU Directive 86/609/EEC, approved by the UCD Ethics Committee and authorized by the Irish Regulatory Authorities.

### Real time PCR

Total RNA was isolated from the colon (8-10-week-old mice) using RNeasy Mini Kit (Qiagen; 74104) and reverse transcribed using the High-Capacity cDNA Reverse Transcription kit (Thermo Fisher Scientific; 4368814). Fast SYBR™ Green Master Mix (Thermo Fisher Scientific; 4385612) was used to perform quantitative real-time PCR on QuantStudio™ 7 Flex system (Thermo Fisher Scientific; 4485701) with gene-specific primers for housekeeping gene *Hprt1* (Forward: 5' GAGGAGTCCTGTTGATGTTGCCAG 3'; Reverse: 5' GGCTGGCCTATAGGCTCATAGTGC 3') and for *Sec1* (Forward: 5' AAGGATCCAAGCAGTGCTCC 3'; Reverse: 5' GGGAAGACCACAAGGGATGG 3').

### Immunofluorescence

5-micron sections of paraffin-embedded colonic tissue fixed in Carnoy's solution (anhydrous ethanol, chloroform, and glacial acetic acid, 6:3:3 ratio) were de-waxed, and antigen was retrieved using 10 mM citric acid (pH 6.0, 15 mins at 95°C). Sections were blocked in 1% BSA and 0.3% Triton-X100 in TBS (Tris-buffered saline, pH 7.6) for 1 hour at room temperature. FITC-conjugated *Ulex Europaeus* agglutinin I (UEA-I) was used at a final concentration of 20 μg/ml (EY labs; FAL-2201-2) to stain tissues for 1h at room temperature. Sections were mounted with ProLong™ Glass Antifade Mountant with NucBlue™ Stain (Thermo Fisher Scientific; P36981). Stained sections were imaged using FV3000 Confocal Laser Scanning Microscope (Olympus). Icy image analysis software (Institute Pasteur) was used for post-acquisition image processing and annotation.

### Mucin purification

12–16-week-old [18] female and male *Cyba*^*nmf333* (*Cyba* mut) and internal *Cyba*^+/+ (WT) control mice on C57BL/6J background were euthanized by cervical dislocation. Intestines were excised, colon and ileum horizontally cut and rinsed carefully with PBS. Colonic (whole colon) and ileal (terminal ileum 10 cm) mucins were isolated and purified as previously published [19]. The procedure was modified by using 8M guanidine hydrochloride (Sigma) for the initial extraction procedure, and for desalting of the samples in the last step by using a Bio-Gel P-6D (Bio-Rad) column.

### *O*-glycan analyses

The samples were immobilized in gels and *N*-glycans were removed from all samples using PNGase F [20]. *O*-glycans were released using microwave-assisted non-reductive β-elimination and labelled with 2-aminobenzamide (2AB) (75%) and with reductive β-elimination (25%) as described in Wilkinson et al [21].

**Microwave-assisted non-reductive β-elimination** was performed as described in Wilkinson et al. [21]. Briefly, the gels containing *O*-glycans were crushed, lyophilized, and resuspended in 250 μL 40% dimethylamine in water with 0.1 g/mL ammonium carbonate. Samples were incubated in a Monowave 450 microwave reactor at 600W (Anton Paar) under 12-minute microwave radiation at 70°C. Samples were then neutralized with 1M HCl, desalted using a Hypersep Hypercarb SPE cartridge, lyophilized and fluorescently labelled with 2AB [22]. The label excess was removed by a Hypersep Diol SPE cartridge [23].

**Reductive β-elimination** and desalting was performed according to Schulz et al [24] and modified for an in-gel release [25].

## Hydrophilic-interaction liquid chromatography-ultra-performance liquid chromatography (HILIC-UPLC)

The 2AB labeled *O*-glycans released by non-reductive β-elimination were resuspended in 20 μL 88% ACN and injected into Glycan BEH Amide, 130Å, 1.7 μm column on an Acquity H-Class HILIC-UPLC with a fluorescence detector (Waters). The glycans were eluted using buffer A (50 mM ammonium formate, pH 4.4) and buffer B (acetonitrile), a column temperature was 40°C. A 30 minute method was used with a gradient of 12% A (0.00–1.47 minutes at 561 μL/min),12–47.6% A (1.47–25.00 minutes at 561 μL/min), 47.6–70% A (25.00–25.60 minutes at 300 μL/min), 70% A (25.60–26.80 minutes at 300 μL/min), 70–12% A (26.80–28.00 minutes at 300 μL/min), 12% A (28.00–30.00 minutes at 561 μL/min) [21]. Fluorescence emission was 420 nm, and excitation 330 nm. External calibration was performed using a 2AB labeled dextran ladder with retention times expressed in glucose units (GUs).

## FLR-HILIC-UPLC-MS

2AB labeled *O*-glycans were desalted using 10 μL normal phase Phytip® Columns (Phynexus Inc.) and resuspended in 10 μL 88% ACN and injected into Glycan BEH Amide column (1.0 x 150 mm, 1.7 μm) on an Acquity H-Class HILIC-UPLC with a fluorescence detector coupled in line with a Xevo G2 Qtof system (Waters). The flow rate was 150 μL/min, and the column temperature 60°C. The glycans were eluted using buffer A (50 mM ammonium formate, pH 4.4) and buffer B (acetonitrile). 30-minute method was used with a gradient: 12% A (0.00–1.00 minutes), 12–47% A (1.00–25.00 minutes), 47–70% A (25.00–25.50 minutes), 70% A (25.50–25.55 minutes at 100 μL/min), 70% A (25.55–26.50 minutes at 100 μL/min), 70%-12% A (26.50–27.00 minutes at 100 μL/min), 12% A (27.00–30.00 minutes at 150 μL/min). Fluorescence emission was 420 nm, and excitation 330 nm. The instrument was operated in negative ion sensitivity mode with a capillary voltage of 1.80 kV for data acquisition. The nitrogen desolvation gas and ion source block temperatures were set at 400°C and 120°C, respectively. The flow rate of desolvation gas was 600 L/h and the cone voltage at 50V. Full scan data for glycans were acquired at m/z range of 450 to 2500. Data analyses were done using MassLynx 4.1 software (Waters).

## Exoglycosidase digestions and their specificities

2AB labeled *O*-glycans were digested using an exoglycosidase enzymes as described by Saldova et al. 2014 [26]. The following enzymes were used: α2–3 sialidase cloned from *Streptococcus pneumoniae* and expressed in *E. coli* (NAN1, digests α2–3 linked non-reducing terminal sialic acids, EC 3.2.1.18), 5 U/mL; α2–3,6,8,9 sialidase cloned from *Arthrobacter ureafaciens* and expressed in *E. coli* (ABS, digests α2–3,6 & 8 linked non-reducing terminal sialic acids, EC 3.2.1.18), 1000 U/mL; β1–3,4 galactosidase cloned from bovine testis and expressed in *Pichia pastoris* (BTG, digests non-reducing terminal galactose β1–3 and β1–4, EC 3.2.1.23), 200 U/

mL; β1–4 galactosidase cloned from *Streptococcus pneumoniae* and expressed in *E. coli* (SPG, digests non-reducing terminal galactose β1–4, EC 3.2.1.23), 80 U/mL; α1–2,3,4,6 fucosidase cloned from bovine kidney and expressed in *E. coli* (BKF, digests non-reducing terminal fucose residues α1–2,6 linked more efficiently than α1–3,4 linked fucose, EC 3.2.1.51), 800 U/mL; β1–2,3,4,6 *N*-acetylglucosaminidase cloned from *Streptococcus pneumoniae* and expressed in *E. coli* (GUH, digests β(1–2,3,4,6) linked GlcNAc, EC 3.2.1.30), 400 U/mL; α1–2,3,6 manno-sidase cloned from *Canavalia ensiformis* (Jack Bean) and expressed in *Pichia pastoris* (JBM, digests non-reducing terminal mannose α1–2,6 linked more efficiently than α1–3 linked, EC 3.2.1.24), 400 U/mL; α1–3,4 fucosidase cloned from the sweet almond tree (*Prunus dulcis*) and expressed in *Pichia pastoris* (AMF, digests non-reducing terminal fucose residues α1–3 & 4 linked, EC 3.2.1.111), 400 mU/mL; β1–3,4,6-*N*-acetylhexosaminidase (β1–4 for GalNAc only) cloned from *Streptomyces plicatus* and overexpressed in *E. coli* (JBH, digests non reducing ter-minal β(1–2,3,4,6) linked GlcNAc and GalNAc residues, EC 3.2.1.52), 800 U/mL; and α1–3,4,6 galactosidase cloned from green coffee bean and expressed in *E. coli* (CBG, digests non-reduc-ing terminal α1–3,4,6 linked galactose, EC 3.2.1.22), 800 U/mL. All enzymes were purchased from New England Biolabs (Hitchin, Herts, U.K.) except for NAN1 which was purchased from Prozyme (San Leandro, CA).

## PGC-LC-MSn

Reductive β-eliminated *O*-glycan alditols were resuspended in 20 µL water and 10% was injected onto a liquid chromatography-electrospray ionization tandem mass spectrometer (LC-ESI-MS) on a 10 cm × 250 µm column packed in-house with 5 µm porous graphitized car-bon particles (Thermo-Hypersil). The flow rate was 5 µL/min, and the column was kept at room temperature. Buffer A was 10 mM ammonium bicarbonate (ABC) and buffer B was 10 mM ABC in 80% acetonitrile. The gradient was 0–45% B (0–46 minutes), washing by 100% B (46–54 minutes), and equilibration with 0% B (54–78 minutes). The samples were analyzed in negative ion mode on a LTQ linear ion trap mass spectrometer (Thermo Electron), with an IonMax standard ESI source with a stainless-steel needle at –3.5 kV. The nebulizer gas was compressed air. The capillary was heated at 270˚C with voltage at –50 kV. Full scan (*m/z* 380–1800, two microscans, maximum ion time 100 ms, full AGC target of 30,000) was followed by data-dependent $MS^2$ or $MS^3$ scans (two microscans, maximum 100 ms, AGC target value of 10,000) with a normalized collision energy of 30%, isolation window of 2.5 units, activation q = 0.25, 30 ms). The MS signal threshold for $MS^2$ and $MS^3$ were 300 and 100 counts, respec-tively. Data acquisition was performed using Xcalibur 2.0.7 software (Thermo Scientific). MS interpretation was aided by comparison to published $MS^2$ spectra made available online on Unicarb DB (unicarb-db.expasy.org).

## Results

Analysis of the colonic mucin *O*-glycosylation profile of ROS-deficient *Cyba* mutant mice will provide additional information on the relation between permanent decrease in intestinal ROS levels and mucin glycans. This study analyzed both colonic and ileal mucins as ROS generation was also strongly downregulated in the terminal ileum of *Cyba* mutant mice [8]. Colonic and ileal mucins were purified from female and male *Cyba^nmf333^* (*Cyba* mut) and internal C57BL/ 6J (WT) control mice. The yield for purified colonic mucin was 10.4 mg (WT, n = 9) and 13.3 mg (*Cyba* mut, n = 16) and for purified ileal mucin 4.1 mg (WT, n = 13) and 6.0 mg (*Cyba* mut, n = 21), respectively. As expected, the yield of ileal and colonic mucin was decreased in *Cyba* mutant mice. As the microbiota of *Cyba* mutant mice was dysbiotic, the expression of *Fut3* was determined. *Sec1* (*Fut3*) expression in colonic tissues of *Cyba* mutant and wildtype

mice was comparable (Fig 1A). Detection of fucosylated oligosaccharides by staining of colonic tissues with *Ulex europaeus* agglutinin 1 (UEA-I) showed slightly less signal deep in the crypts of *Cyba* mutant mice, but the pronounced change in the thickness and density of MUC2 mucin between wildtype and mutant mice did not permit any quantitative assessment (Fig 1B).

After extraction and purification of colonic and ileal mucins from WT and *Cyba* mut mice detailed *O*-glycan analyses were performed on the pooled WT and *Cyba* mut samples by

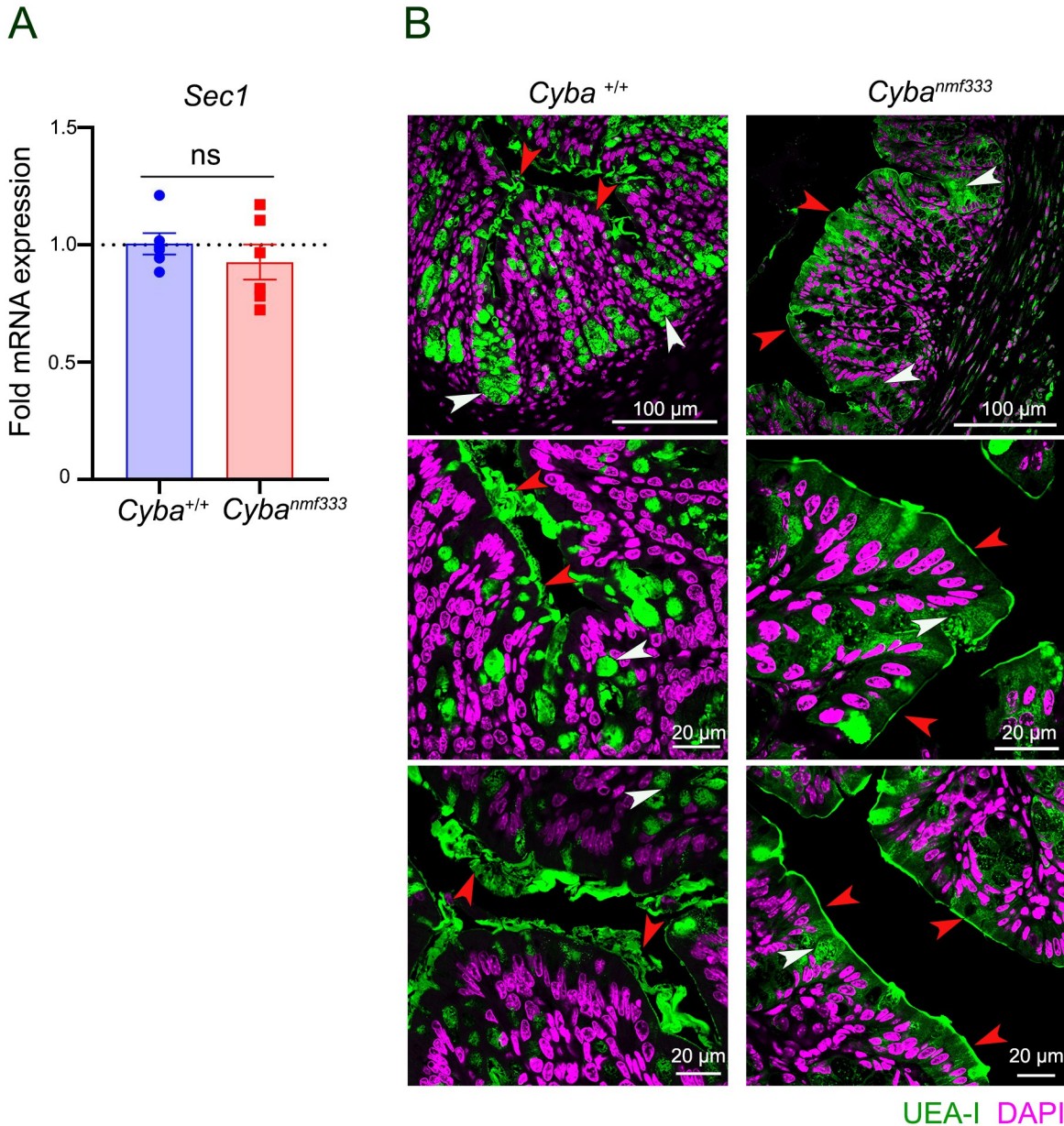

**Fig 1. Comparison of *Sec1* fucosyltransferase expression and the presence of fucosylated oligosaccharides in wildtype and *Cyba* mutant mice.** A) Relative expression of *Sec1* (*Fut3*) in wildtype (*Cyba*$^{+/+}$, WT) and *Cyba*$^{nmf333}$ mouse colon (n = 5). B) Representative immunofluorescence images of UEA-I staining in the colon of wildtype (*Cyba*$^{+/+}$, WT) and *Cyba*$^{nmf333}$ mice (n = 4/group; 12 images each). UEA-I in green, DNA in magenta, arrows indicate goblet cells (white) and mucus layer (red).

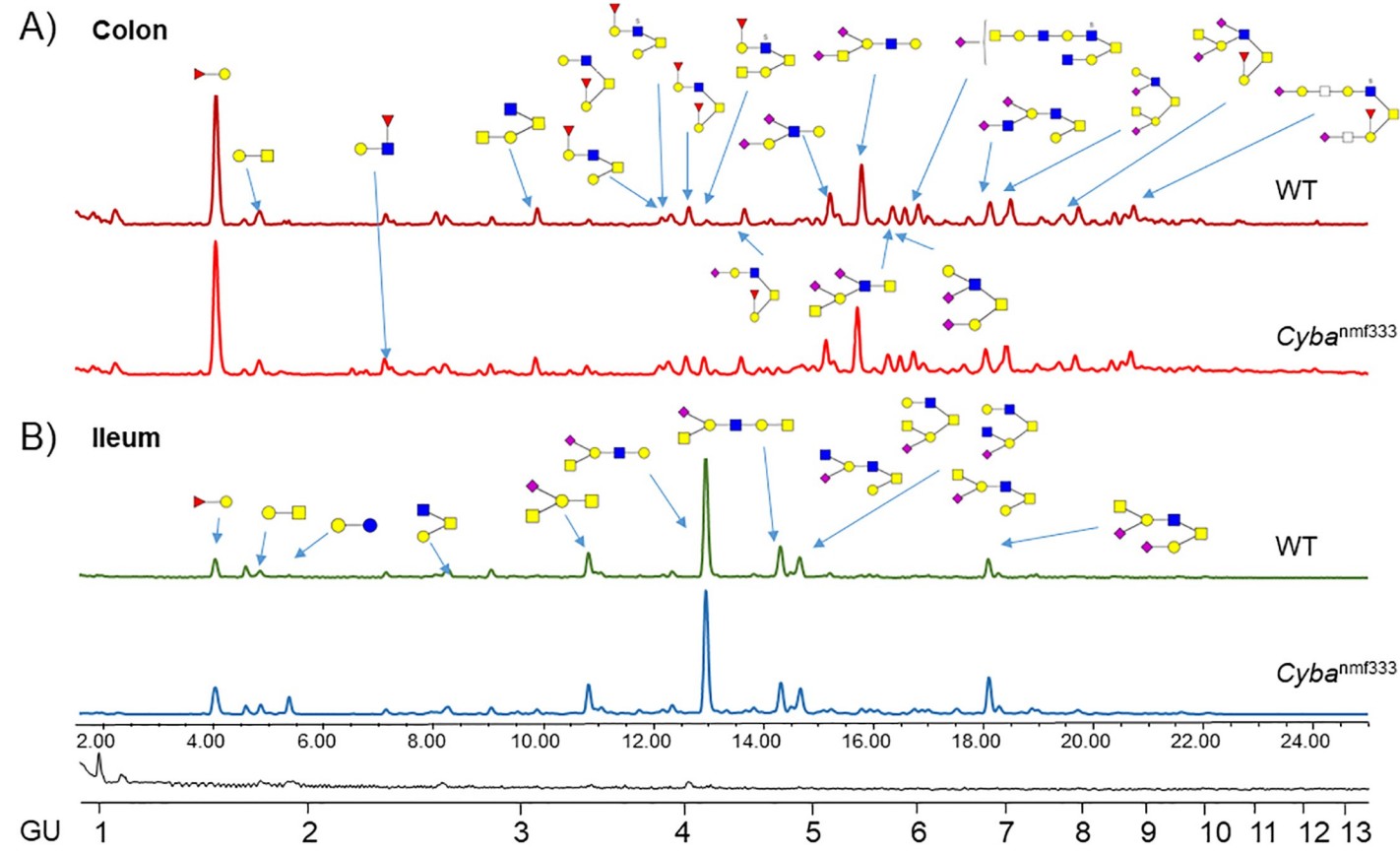

**Fig 2. Typical/pooled HILIC-UPLC profile of A) colonic and B) ileal *O*-glycans derived from WT and *Cyba^nmf333* mice.** Main *O*-glycans are labelled. Detailed *O*-glycan composition of these samples is listed in Table 1.

fluorescence liquid chromatography (FLR-HILIC-UPLC) in combination with exoglycosidase digestions for quantification and mass spectrometry (PGC-LC-MS^n) for sequence characterization. This method shows good reproducibility with most chromatographic peaks with CV below 15–20% for mucins [21,27]. However, in our sample performing ten replicates on pooled mucin glycan samples from colon and ileum, integrating 127 peaks, the reproducibility was found to be lower (S1 Table), and therefore we focused on identifying glycan features. Typical HILIC-UPLC profiles from *Cyba* mut and WT mucins derived from colon and ileum are shown in Fig 2. The glycan profiles were separated into 123 peaks in colon and into 109 peaks in ileum HILIC-UPLC chromatograms (S1 Fig). All characterized *O*-glycans from these samples are displayed in Table 1A and 1B. Detailed glycan assignments using both methods are shown in S2 and S3 Tables. We used exoglycosidase enzyme arrays to identify specific linkages and digest most of the glycans to the cores. Namely, using sialidases ABS and NAN1 identified sialic acid linkages, galactosidases BTG and SPG galactoses linkages, fucosidases BKF and AMF fucose linkages, and GUH and JBH revealed the presence of *N*-Acetylglucosamine (GlcNAc) or *N*-Acetylgalactosamine (GalNAc) (S3 Table). Exoglycosidase enzymes used, and their specificities are described in the Materials and Methods section.

Overall, the HILIC-UPLC method showed 24–35% of peeling and 16–24% undigestible/contaminating glycans in the profiles (S3 Table). In murine colon core 1, 2, 3, and 4 glycans were detected with most abundance of core 2, and core 3 and 4 being the least abundant (Fig

**Table 1.** A. List of all *O*-linked glycans from colonic mucins from WT and *Cyba* mut mice. B. List of all *O*-linked glycans from ileal mucins from WT and *Cyba* mut mice.

| Name[1] | Tentative sequence | GU |
|---|---|---|
| 368-1PH | Fucα1-2Gal | 1.60 |
| 384–1 | Galβ1-3GalNAc | 1.79 |
| 384-2PH | GlcNacβ1-6Gal | 1.82 |
| 343-1PH | Galβ1-3Glc | 1.97 |
| 530–1 | Fucα1-2Galβ1-3GalNAc | 2.26 |
| 530-2PH | Galβ1-3(Fucα1–2)GlcNAc | 2.40 |
| 546-1PH | Galβ1-3GlcNAcβ1-6Gal | 2.62 |
| 587–1 | Galβ1–3[GlcNAcβ1–6]GalNAc | 2.68 |
| 667–1 | Galβ1–3[**SO3-GlcNAcβ1–6**]GalNAc | 2.90 |
| 675–1 | NeuAcα2-3Galβ1-3GalNAc | 2.99 |
| 733–1 | Fucα1-2Galβ1–3[GlcNAcβ1–6]GalNAc | 2.95 |
| 749–1 | Galβ1-3(Galβ1-3GlcNAcβ1–6)GalNAc | 3.43 |
| 749–2 | Galβ1-3(Galβ1-4GlcNAcβ1–6)GalNAc | 3.48 |
| 790–3 | GalNAcβ1-6Galβ1-3(GlcNAc1-6)GalNAc | 3.20 |
| 813–1 | Fucα1-2Galβ1-3(**SO3-GlcNAcβ1-6**)GalNAc | 3.33 |
| 878–1 | Galβ1-3(NeuAcα2–6)GlcNAcβ1-3GalNAc | 3.52 |
| 895–1 | Galβ1–3[Fucα1-2Galβ1-3GlcNAcβ1–6]GalNAc | 3.89 |
| 895–2 | Galβ1–3[Fucα1-2Galβ1-4GlcNAcβ1–6]GalNAc | |
| 895–3 | Fucα1-2Galβ1–3[Galβ1-3/4GlcNAcβ1–6]GalNAc | 3.89 |
| 975–1 | Galβ1–3[Fucα1-2Galβ1-3/4**SO3-GlcNAcβ1–6**]GalNAc | 3.96 |
| 1041–1 | Fucα1-2Galβ1–3[Fucα1-2Galβ1-3GlcNAcβ1–6]GalNAc | 4.08 |
| 1041–2 | Fucα1-2Galβ1–3[Fucα1-2Galβ1-4GlcNAcβ1–6]GalNAc | 4.28 |
| 1121–1 | Fucα1-2Galβ1–3[Fucα1-2Galβ1-3**SO3-GlcNAcβ1–6**]GalNAc | 4.32 |
| 1121–2 | Fucα1-2Galβ1–3[Fucα1-2Galβ1-4**SO3-GlcNAcβ1–6**]GalNAc | 4.42 |
| 1169–1 | NeuAc2-3GalNAcβ1-4(NeuAcα2–6)Galβ1-3GalNAc/GlcNAc | 5.00 |
| 1169–2 | NeuAc2-3/6GlcNAc/GalNAcβ1-3/4/6(NeuAcα2–6)Galβ1-3GalNAc/GlcNAc | 5.09 |
| 1128-1PH | NeuAc2-3Galβ1-3(NeuAcα2–6)GlcNAcβ1-6Gal | 5.22 |
| 1178–1 | GalNAcβ1-4Galβ1–3[Fucα1-2Galβ1-3/4SO3-GlcNAcβ1–6]GalNAc | 4.20 |
| 1186–1 | Fucα1-2Galβ1–3[NeuAcα2–6Galβ1-3GlcNAcβ1–6]GalNAc | 4.49 |
| 1243–1 | Galβ1–3[GalNAcβ1-4NeuAcα2–6Galβ1-3/4GlcNAcβ1–6]GalNAc | 4.89 |
| 1243–1 | Galβ1–3[GalNAβ1-4Galβ1-3(NeuAcα2-3/6)GlcNAcβ1–6]GalNAc | 5.28 |
| 1266–1 | Galβ1–3[NeuAcα2-3/6Galβ1-3/4(Fucα1–2)**SO3-GlcNAcβ1–6**]GalNAc | 5.28 |
| 1372–1 | GlcNAc/GalNAcβ1-4/6(NeuAcα2–3)Galβ1–3[NeuAc2-3/6GlcNAcβ1–6]GalNAc | 5.28 |
| 1131–1 | NeuAcα2–6Galβ1–3[Galβ1-3(NeuAcα2–6)GlcNAcβ1–6]GalNAc | 5.95 |
| | NeuAcα2-3/6Galβ1–3[Galβ1-4(NeuAcα2-3/6)GlcNAcβ1–6]GalNAc | 6.21 |
| 1331-1P | NeuAcα2–6GalNAcβ1-4(NeuAcα2–6)Galβ1-3GlcNAcβ1-6Gal | 5.51 |
| 1372–2 | GalNAcβ1-4(NeuAcα2-3/6)Galβ1-3/4(NeuAcα2-3/6)GlcNAcβ1-3GalNAc | 5.83 |
| 1389–1 | Fucα1-2Galβ1–3[GlcNAcβ1-3(NeuAcα2–6)Galβ1-3GlcNAcβ1–6]GalNAc | 5.69 |
| 1469–1 | NeuAcα2-3/6(Fucα1–2)Galβ1–3[GalNAc/GlcNAcβ1-3/4/6Galβ1-3/4SO3-GlcNAcβ1–6]GalNAc | 5.38 |
| 1477–1 | Fucα1-2Galβ1–3[NeuAcα2–6Galβ1-3(NeuAcα2-3/6)GlcNAcβ1–6]GalNAc | 6.67 |
| 1535–1 | Galβ1–3[GalNAβ1-4(NeuAcα2–3)Galβ1-3(NeuAcα2-3/6)GlcNAcβ1–6]GalNAc | 6.94 |
| 1535–1 | Galβ1–3[NeuAcα2-6GlcNAcβ1-6(NeuAcα2–6)Galβ1-3/4GlcNAcβ1–6]GalNAc | 6.94 |
| 1535–1 | GalNAcβ1-4(NeuAcα2–6)Galβ1–3[Galβ1-3(NeuAcα2-3/6)GlcNAcβ1–6]GalNAc | 7.21 |

(*Continued*)

**Table 1.** (Continued)

| 1681–1 | Fucα1-2Galβ1–3[GlcNAc/GalNAcβ1-3/4(NeuAcα2–6)Galβ1-3/4(NeuAcα2-3/6)GlcNAcβ1–6]GalNAc (4x dif isomers- dif in GalNAc/GlcNAc and galactose linkage) | 7.62/ 7.91/ 8.14 |
|---|---|---|
| 1843–1 | NeuAcα2-3/6-, Fucα-GalNAcβ1-4Galβ1-3/4GlcNAc/GalNAcβ1-3/4/6Galβ1-3/4GlcNAcβ1-6Gal | 8.57 |
| 1843–2 | | 8.86 |
| 1892–1 | NeuAcα2-3/6-GlcNAcβ1-3/6Galβ1–3[GalNAcβ1-4Galβ1-3/4GlcNAcβ1-3/6Galβ1-3/4SO3-GlcNAcβ1–6]GalNAc | 6.10 |
| 2126–1 | NeuAcα2-3/6GalNAc/GlcNAcβ1-3/4/6(Fucα1–2)Galβ1–3[NeuAcα2-3/6)Galβ1-3GalNAc/GlcNAcβ1-3/4/6Galβ1-3SO3-GlcNAcβ1–6]GalNAc | 9.00 |
| | Fucα1-2Galβ1-3(GalNAc/GlcNAcβ1-3/4[NeuAcα2-6Galβ1-3/4GalNAc/GlcNAcβ1-4/6(NeuAcα2-3/6)Galβ1-3/4SO3-GlcNAcβ1–6]GalNAc | 9.00 |
| | Fucα1-2Galβ1-3[GalNAc/GlcNAcβ1-4/6(NeuAcα2–3)Galβ1-3/4GalNAc/GlcNAcβ1-3/4/6Galβ1-3/4(NeuAcα2-3/6)SO3-GlcNAcβ1–6]GalNAc | 9.30 |
| 2126–1 | NeuAcα2-3/6GalNAc/GlcNAcβ1-3/4/6(Fucα1–2)Galβ1–3[NeuAcα2-3/6Galβ1-3GalNAc/GlcNAcβ1-3/4/6Galβ1-4SO3-GlcNAcβ1–6]GalNAc | 9.99 |
| 2249–1 | (NeuAcα2-3/6)2-Galβ1-3/4(Fucα1–2)GlcNAcβ1-3/6[GalNAcβ1-4Galβ1-3/4GlcNAcβ1-3/6Galβ1-3/4GlcNAcβ1-3/6]GalNAc | 10.90 |

| Name[1] | Tentative sequence | GU |
|---|---|---|
| 368-1PH | Fucα1-2Gal | 1.59 |
| 384–1 | Galβ1-3GalNAc | 1.80 |
| 425–1 | GlcNAcβ1-6GalNAc | 1.82 |
| 505–1 | SO3-GlcNAcβ1-3GalNAc | |
| 343-1PH | Galβ1-3Glc | 1.96 |
| 530–1 | Fucα1-2Galβ1-3GalNAc | 2.26 |
| 472-P | NeuAcα2-3Gal | 2.40 |
| 505-P | Galβ1-3Galβ1-3Gal | 2.62 |
| 587–1 | Galβ1–3[GlcNAcβ1–6]GalNAc | 2.69 |
| 667–1 | Galβ1–3[**SO3-GlcNAcβ1–6**]GalNAc | 2.91 |
| 675–1 | NeuAcα2-3Galβ1-3GalNAc | 3.01 |
| 749-PH | GalNAcβ1-4Galβ1-3/4GlcNAcβ1-6Gal | 3.23 |
| 813–1 | Fucα1-2Galβ1–3[**SO3-GlcNAcβ1–6**]GalNAc | 3.34 |
| 878–1 | GalNAcβ1–4[NeuAc2-3/6]Galβ1-3GalNAc | 3.44 |
| 749–1 | Galβ1–3[Galβ1-3GlcNAcβ1–6]GalNAc | 3.36 |
| 749–2 | Galβ1–3[Galβ1-4GlcNAcβ1–6]GalNAc | 3.48 |
| 829–1 | Galβ1–3[Galβ1-4**SO3-GlcNAcβ1–6**]GalNAc | 3.65 |
| 829–2 | Galβ1–3[Galβ1-3**SO3-GlcNAcβ1–6**]GalNAc | |
| 878-PH | GalNAcβ1-4(NeuAcα2–6)Galβ1-3GlcNAc | 3.75 |
| 952-1H | GalNAcβ1-4Galβ1-4GlcNAcβ1-3/6Galβ1-3GalNAc | 3.91 |
| 952-2H | GalNAcβ1-4Galβ1-3GlcNAcβ1-3/6Galβ1-3GalNAc | 3.97 |
| 952-3H | Galβ1–3[GalNAc/GlcNAcβ1-3/4/6Galβ1-4GlcNAcβ1–6]GalNAc | 4.10 |
| 952-4H | Galβ1–3[GalNAc/GlcNAcβ1-3/4/6Galβ1-3GlcNAcβ1–6]GalNAc | 4.14 |
| 1040-P | GalNAcβ1-4(NeuAcα2–6)Galβ1-3/4GlcNAcβ1-6Gal | 4.21 |
| 1040–1 | NeuAcα2-3/6Galβ1–3[Galβ1-3/4GlcNAcβ1–6]GalNAc | 4.35 |
| 1040–2 | Galβ1–3[NeuAcα2-3Galβ1-4GlcNAcβ1–6]GalNAc | 4.52 |
| 1040–3 | Galβ1–3[NeuAcα2-6Galβ1-4GlcNAcβ1–6]GalNAc | 4.58 |
| 1243–1 | GalNAcβ1-4(NeuAcα2–6)Galβ1-3GlcNAcβ1-3/6Galβ1-3GalNAc | 4.81 |
| 1243–2 | Galβ1–3[GalNAc/GlcNAcβ1-3/4(NeuAcα2–6)Galβ1-4GlcNAcβ1–6]GalNAc | 4.97 |
| 1243–3 | NeuAcα2-3/6(GalNAc/GlcNAcβ1-3/4/6)Galβ1–3[Galβ1-4GlcNAcβ1–6]GalNAc | |
| 1243–4 | NeuAcα2-3/6Galβ1-3(GalNAcβ1-4Galβ1-3/4GlcNAcβ1-6GalNAc | 6.18 |
| 1155–1 | GlcNAcβ1-3/6Galβ1–3[GalNAcβ1-4Galβ1-4GlcNAcβ1–6]GalNAc | 5.38 |

(*Continued*)

**Table 1.** (Continued)

| | | |
|---|---|---|
| 1389–1 | GalNAcβ1-4(NeuAcα2-3/6)Galβ1–3[Galβ1-4(Fucα1–2)GlcNAcβ1-6GalNAc | **5.62** |
| 1389–2 | [NeuAcα2-3/6] +(GlcNAcβ1-3/6(Fucα1–2)Galβ1–3[Galβ1-4GlcNAcβ1–6]GalNAc | |
| 1446–1 | GlcNAcβ1-3/6Galβ1–3[GalNAcβ1-4(Neu5Acα2–6)Galβ1-3/4GlcNAcβ1–6]GalNAc | **6.10** |
| 1446–2 | GalNAcβ1-4(NeuAcα2–3)Galβ1–3[GalNAcβ1-4Galβ1-4GlcNAcβ1–6]GalNAc | **6.58** |
| 1446–3 | GalNAcβ1-4(NeuAcα2-3/6)Galβ1-3(GalNAcβ1-4Galβ1-3GlcNAcβ1–6)GalNAc | **6.74** |
| 1535–1 | NeuAcα2-3/6Galβ1–3[GalNAcβ1-4(NeuAcα2–6)Galβ1-3/4GlcNAcβ1–6]GalNAc | **6.96** |
| 1608–1 | GalNAcβ1-4(NeuAcα2–6)Galβ1-3/4GlcNAcβ1-3/6Galβ1-3/4GlcNAcβ1-3/6Galβ1-3GalNAc | **7.09** |
| 1811–1 | [GlcNAcβ1–6]GalNAcβ1-4Galβ1-3/4GlcNAcβ1-3/6Galβ1-3/4(NeuAcα2-3/6)GlcNAcβ1-3/6Galβ1-3GalNAc | **7.38** |
| 1738–1 | GalNAcβ1-4(NeuAcα2-3/6)Galβ1–3[GalNAcβ1-4(NeuAcα2–6)Galβ1-3/4GlcNAcβ1–6]GalNAc | **7.51/ 7.59** |
| 2103–1 | GalNAcβ1-4(NeuAcα2–6)Galβ1-3/4GlcNAcβ1-3/6Galβ1–3[GlcNAcβ1-3/6(NeuAcα2-3/6)Galβ1-4GlcNAcβ1–6]GalNAc | **8.14** |

[1] P = peeling, PH = peeling only found in HILIC (not MS), H = glycan found only in HILIC.
Novel structures are highlighted.

2A, Tables 1A and S3A). In ileum, core 1, 2 and 3(6) glycans were present with core 2 being the most abundant followed by substantial abundance of core 1, and core 3(6) being the least abundant (Fig 2B, Tables 1B and S3B). We have identified five novel glycans in ileum not previously published: GlcNAcβ1-3/6Galβ1–3[GalNAcβ1-4(Neu5Acα2–6)Galβ1-3/4GlcNAcβ1–6] GalNAc (1446–1), GalNAcβ1-4(NeuAcα2–3)Galβ1–3[GalNAcβ1-4Galβ1-4GlcNAcβ1–6]Gal-NAc (1446–2), GalNAcβ1-4(NeuAcα2-3/6)Galβ1-3(GalNAcβ1-4Galβ1-3GlcNAcβ1–6)Gal-NAc (1446–3), GalNAcβ1-4(NeuAcα2–6)Galβ1-3/4GlcNAcβ1-3/6Galβ1-3/4GlcNAcβ1-3/ 6Galβ1-3GalNAc (1608–1) and [GlcNAcβ1–6]GalNAcβ1-4Galβ1-3/4GlcNAcβ1-3/6Galβ1-3/4 (NeuAcα2-3/6)GlcNAcβ1-3/6Galβ1-3GalNAc (1811–1) (Fig 3, S2B Table).

When comparing colonic and ileal *Cyba* mut with WT samples, the same glycans were identified in both pools, namely core 1–4 glycans both fucosylated, sialylated and/or sulfated (Tables 1 and S2). Quantitative analyses were done using both PGC-LC-MS method and HILIC-UPLC in combination with the exoglycosidase digestion method. PGC-LC-MS quantified glycans as % area from all identified glycans, and HILIC-UPLC similarly as % area from all identified glycans as well as % area from all released glycans. When comparing colonic and ileal *Cyba* mut with WT mucins, no major changes were found when taking into account all methods. In colon, there was an indication of a minor increase in glycans with terminal Hex-NAc, core 2 glycans with Fuc-Gal- on C3 branch and glycans containing (sialyl)Lewis x epitope, and a decrease in core 3 glycans (Fig 3, S2A and S4 Tables). In contrast, in ileal *Cyba* mut structures an increase in sialylation and a decrease in sulfated glycans was indicated (Fig 4, S3A and S4 Tables).

Comparison of colonic and ileal glycans showed in WT and *Cyba* mut colon a pronounced increase of core 2, core 4, and fucosylated and sulfated glycans as well as a decrease in core 1 and sialylated glycans. This indicates differences in glycosylation in distinct areas of the murine intestine (Fig 5, S4 Table), in line with previous publications on mouse ileum/colon glycosylation [30,31].

## Discussion

*Cyba* mutant mice are highly susceptible to chemically induced colitis due to their compromised epithelial barrier and inability to mount an efficient microbicidal host response [8]. At

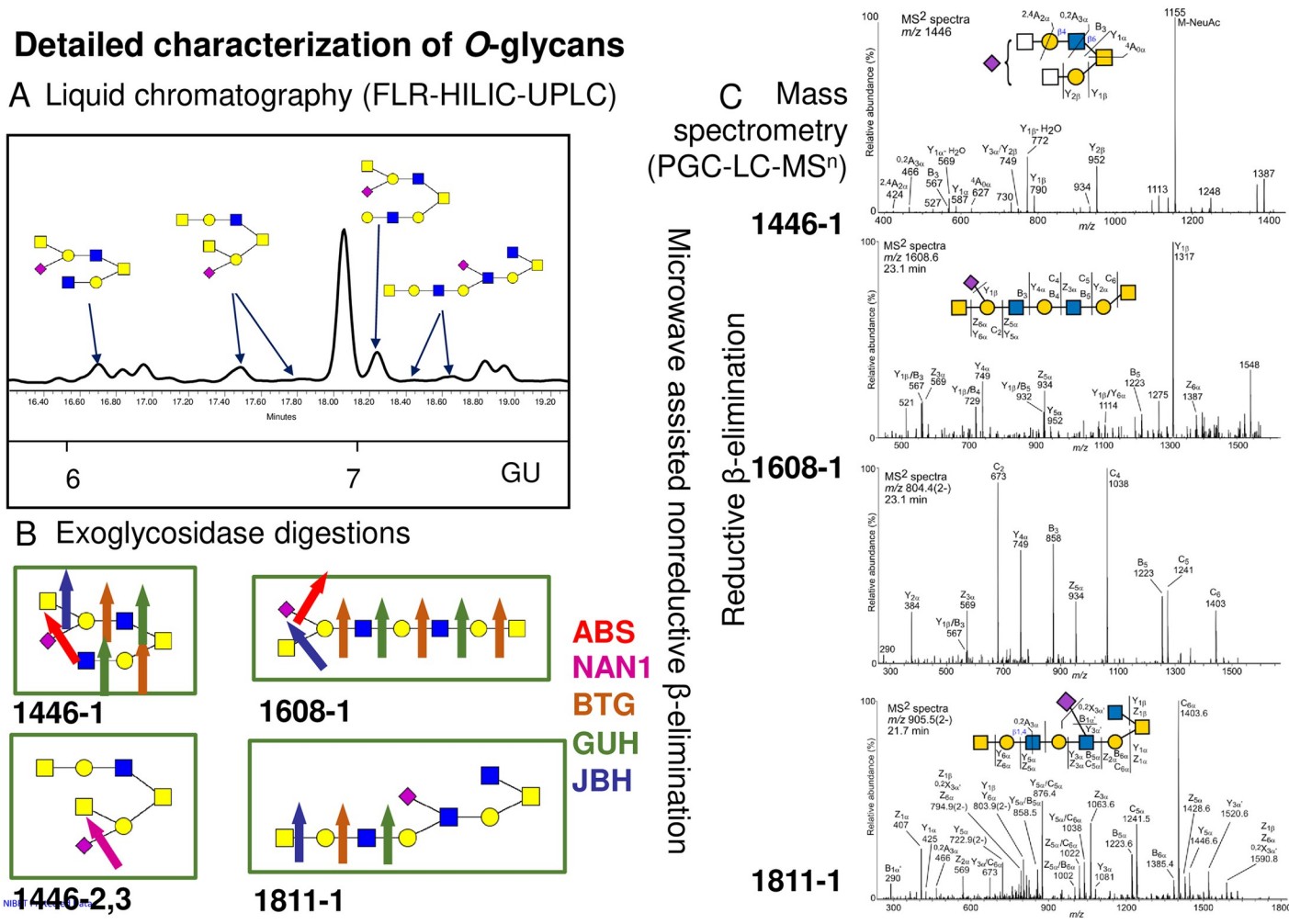

**Fig 3. Novel glycans found in ileal samples and their detailed characterization using combination of FLR-HILIC-UPLC, exoglycosidase digestions and PGC-LC-MSn. A)** Part of HILIC-UPLC chromatogram with novel glycans pictured. **B)** Novel glycans digested by exoglycosidase enzymes, namely α2-3/6/8 sialidase (ABS), α2-3/8 sialidase (NAN1), β1-3/4/6 galactosidase (BTG), β1-2/4/6 hexosamindase (GUH) and β1-2/3/4/6 hexosaminase (JBH). Detailed assignments are in S3B Table. **C) 1446**. MS/MS spectra of a reduced and nonderivatized sialylated *O*-glycan '1446–1' (S2B Table) eluting at 20.6 min and detected at m/z 1446.5 ([M-H]- precursor ion) in ileal mucus of WT control mice. **1608**. MS/MS spectra of the mono- and doubly charged precursor ions of a reduced and nonderivatized sialylated *O*-glycan '1608–1' (S2B Table) eluting at 23.1 min and purified from ileal mucus of *Cyba^nmf333^* (*Cyba* mut) mice. Top panel: m/z 1608.6 ([M-H]- precursor ion); Bottom panel: m/z 804.4 ([M-2H]2- precursor ion). **1811**. MS/MS spectra of a reduced and nonderivatized sialylated *O*-glycan '1811–1' (S2B Table) eluting at 21.7 min and detected at m/z 905.5 ([M-2H]2- precursor ion) in ileal mucus of WT control mice. The analyses were performed using porous graphitized carbon liquid chromatography and low-resolution mass spectrometry kept in the negative ion mode. Proposed structures are shown using SNFG [28] and glycan fragmentation is as defined by Domon and Costello [29].

homeostasis the loss of superoxide generation by Nox1-3 resulted in a thin or absent dense mucus layer due to altered mucus secretion with direct contact of microbiota with the crypt epithelium and pronounced bacterial dysbiosis with increased abundance of mucus-associated *Mucispirillum schaedleri*. As the microbiota of *Cyba* mutant mice is dysbiotic, fucosyltransferase expression was measured. In the earlier study by Aviello and coworkers the expression of the fucosyltransferase *Fut1* was not altered while expression of *Fut2* was decreased by 30–40% [8]. We found comparable *Sec1* (*Fut3*) expression in colonic tissues of *Cyba* mutant and wild-type mice in this study (Fig 1A).

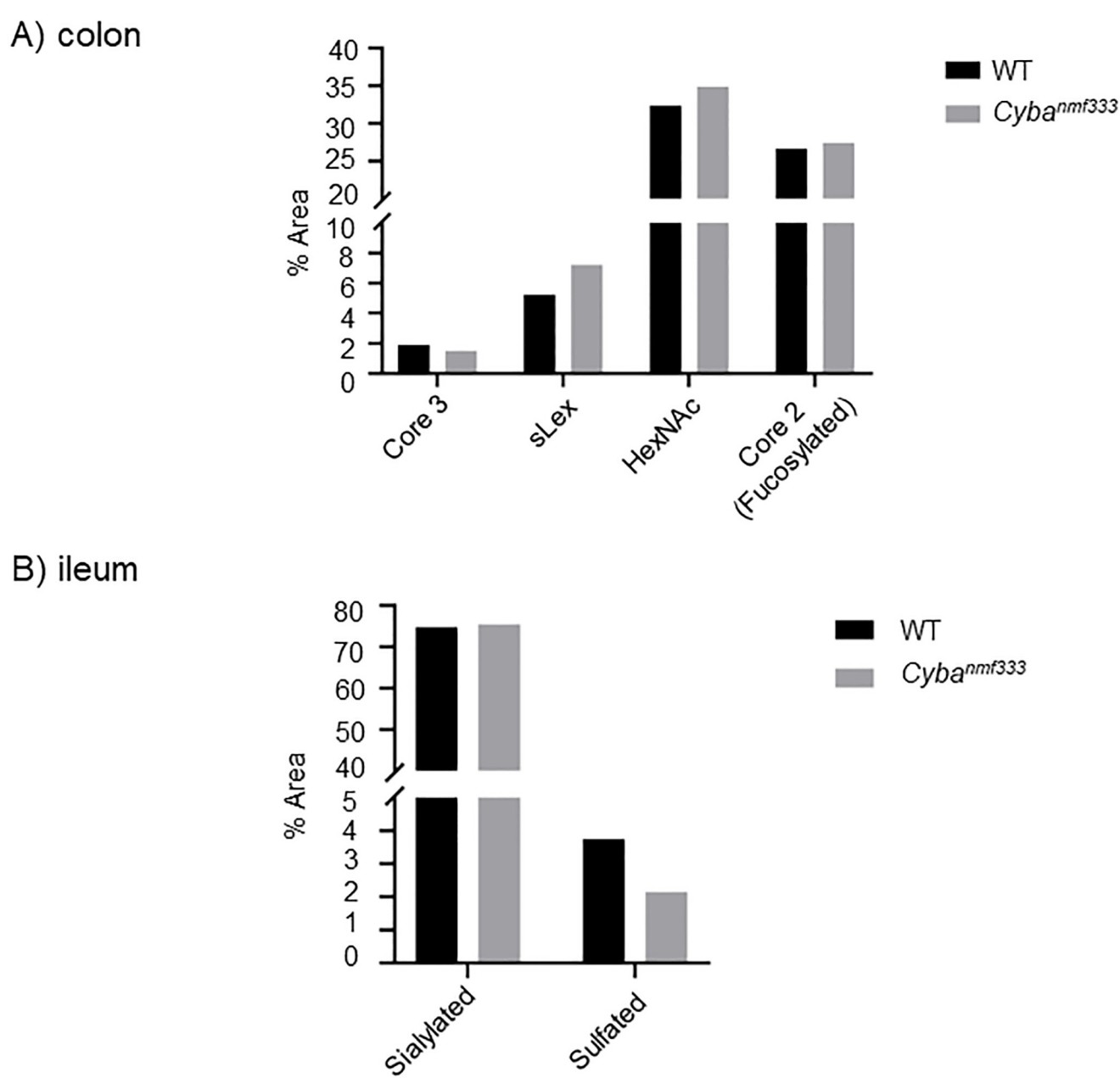

**Fig 4. Comparison of *O*-glycan features in wildtype (WT) and *Cyba^nmf333^* mouse colon and ileum.** Glycan features in A) colon and B) ileum that show differences in WT and *Cyba^nmf333^* mice using all methods. Core 3 group in ileum contains a low amount of core 6 glycans. Graphs are based on quantitation by HILIC-UPLC method.

Next, we performed comprehensive glycan analysis, consisting of a combination of liquid chromatography, exoglycosidase digestions and mass spectrometry which enabled us to characterize the glycans with more depth as previously published studies, especially in relation to linkages and specific residues which our method could separate. We found five additional novel glycans in the ileum (Fig 3 and S2B Table). These methods are complementary and provide the most comprehensive glycan assignments by using recently developed microwave assisted nonreductive β-elimination for the HILIC preparation which improves reproducibility

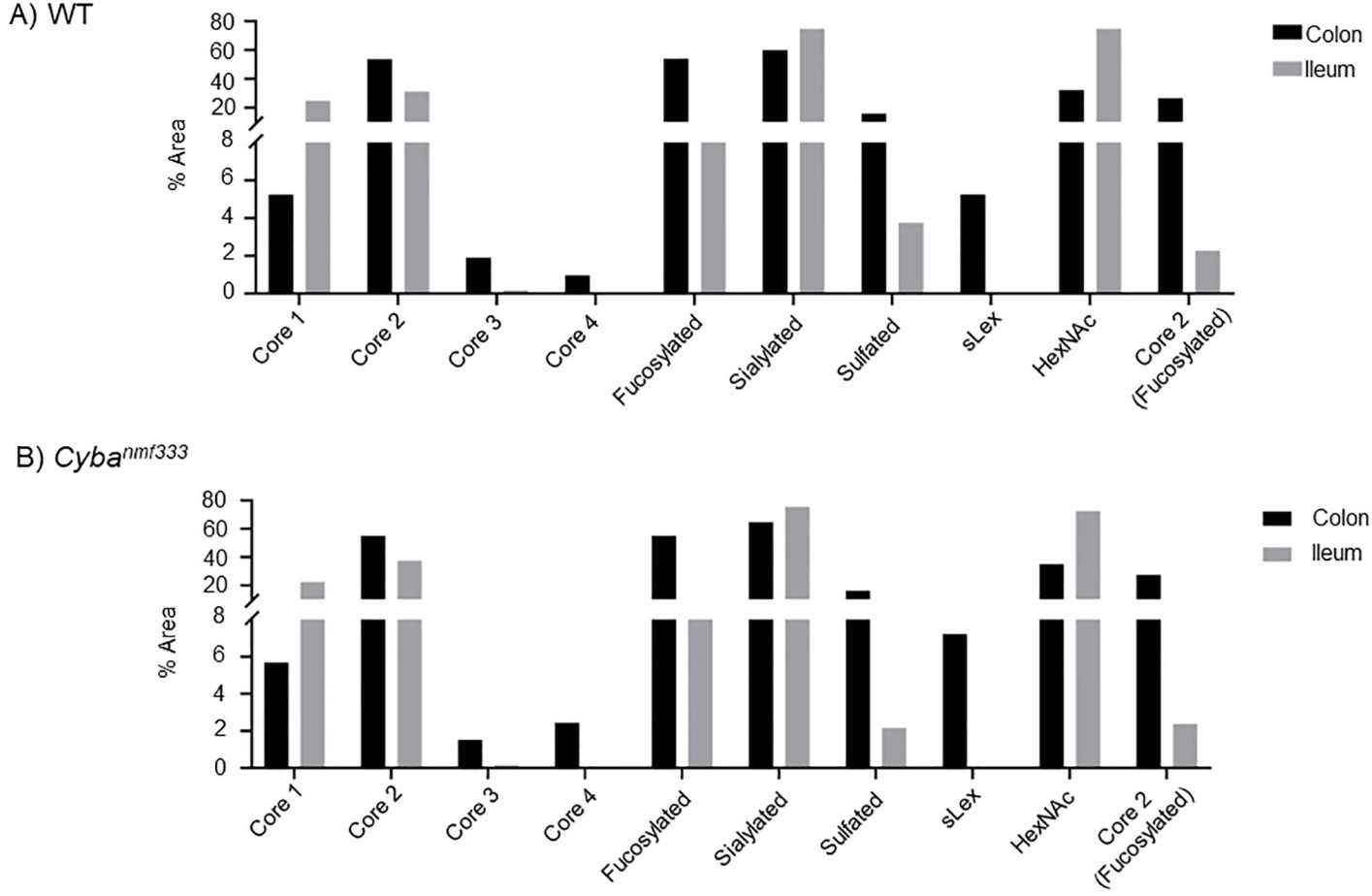

**Fig 5. Comparison of *O*-glycosylation in colon versus ileum is different in wildtype (WT) and *Cyba^nmf333^* mice.** Glycan features in colonic vs ileal mucins in A) WT and B) *Cyba^nmf333^* mice. Graphs are based on quantitation by the HILIC-UPLC method.

of *O*-glycan analyses [21]. As we found slightly lower reproducibility in our glycan peaks, we focused on the comparison of glycan features rather than individual glycans.

The importance of the mucus layer in providing protection for the host's epithelial barrier and as niche for colonization and nutrition of commensal facultative anaerobe and oxygen tolerant bacteria prompted several mucin *O*-glycome studies in humans and mice to date. Mucin-type *O*-glycosylation in various segments of the colon and small intestine varies considerably, as expected by differential expression of glycosyltransferases, region-specific physicochemical changes in mucus layer structure and host physiology. Differences between humans and mice exist in the overall abundance of *O*-glycan core structures and in the prevalence of modifications such as fucosylation versus sialylation in the small intestine [31–33]. In conventional C57BL/6N mice regiospecific distribution of core 2 and core 1 glycans correlates well with our analyses in conventional *Cyba^+/+^* (WT) and *Cyba* mutant mice on a C57BL/6J background [30,31].

In colon core 1, 2, 3, and 4 glycans were detected, whereas in ileum core 1, 2 and 3(6) glycans were present (Figs 2 and 4, Tables 1 and S3). Comparison of ROS-deficient *Cyba* mutant mouse mucins with wildtype mouse mucins did not show pronounced differences in ileal glycan structures, although changes in mucus secretion or *in situ* preservation were evident by

low mucin yields. When the individual glycans were pooled into the features, several differences between *Cyba* mutant and control mice were noted. Host ROS deficiency resulted in an increase in the levels of glycans with terminal HexNAc, core 2 glycans with Fuc-Gal- on C3 branch and glycans containing (sialyl)Lewis x epitope. A trend to decreased levels of core 3 glycans in *Cyba* mutant mouse colon (Fig 3, S4 Table) was also observed. Further, when comparing ileal mucins in *Cyba* mutant with control mice, an increase in sialylated and a decrease in sulfated glycans was detected (Fig 3, S4 Table).

Changes in mucin *O*-glycans have been reported in IBD patients and in mice in association with spontaneous or heightened susceptibility to colitis. In IBD patients MUC2 protein, but not mRNA, was found to be increased, hypoglycosylated and/or poorly sulfated [34]. It was suggested that glycotransferases in the goblet cell secretory apparatus or mucin glycan degrading enzymes, either mammalian or bacterial, may be responsible for the distinct glycan profiles [35]. MUC2 protein of patients with active UC was composed of smaller glycans than in UC remission or in non-IBD individuals, suggesting that altered glycosylation in UC is closely associated with clinical severity of disease and may return to normal during remission [35]. Sialylation was increased in UC, suggesting that inflammation affects the glycosylation of mucins.

An increase in sialyl Lewis x epitope, upregulated in inflammation, was also observed in IBD in line with our findings [15]. Indeed, the glycosylation changes observed in IBD were found to coincide with inflammation in acute infections [36]. An upregulation of sialyltransferases results in an increase in the STn antigen (sialosyl-GalNAc-S/T) on MUC2, which leads to smaller glycans, less complex glycosylation and a subsequent decrease in mucin sulfation [35,37]. Mice lacking the sulfate transporter NaS1 have reduced intestinal mucin sulfation and increased susceptibility to bacterial infection and DSS-induced colitis [38]. Consistent with these findings, mice lacking *N*-acetylglucosamine-6-*O*-sulfotransferase, which catalyzes the transfer of sulfates to mucins, showed decreased mucin sulfation, increased inflammation and increased susceptibility to DSS-induced colitis [39]. Mice lacking both intestinal core 1- and core 3- derived *O*-glycans developed spontaneous colitis in both distal and proximal colon [13]. Absence of core 3 *O*-glycans in mice lacking the β1,3–*N*-acetylglucosaminyl-transferase enzyme resulted in decreased MUC2 synthesis and a diminished mucus barrier, leading to increased susceptibility to dextran sulfate sodium (DSS)-induced colitis and development of colorectal tumors [40]. Similar results were reported in mice lacking the core 2 β 1,6-*N*-acetyl-glucosaminyl-transferase enzyme [41]. While mucins have not been analyzed in pediatric IBD patients with decreased oxidase activity due to mutational NOX1, NOX2 complex or DUOX2 inactivation, we show here that mucins of *Cyba* mutant mice, a murine model for a subset of CGD patients, displayed more sialyation and decreased sulfation at homeostasis, which together with decreased mucus secretion reflects an inflammation-susceptible environment.

Deviations in post-translational modifications of mucins such as glycosylation and sulfation have an impact on mucin degradation [35,42,43]. Modified mucins will select for particular bacterial species that can attach, colonize and thrive better in these conditions, thereby displacing other commensals and reducing microbiota diversity. For example, the increase in mucus-degrading bacteria in IBD such as *Ruminococcus* spp. may degrade the mucus layer and increase the direct exposure of colon epithelial cells to microbiota [44,45]. In accord, the dysbiotic microbiota of mice with decreased ROS generation such as *Cyba*[nmf333] mice or *Ncf1*[-/-] mice (p47[phox] knockout, i.e. Nox2 complex inactivation) harbored an abundance of mucus-associated *Mucispirillum schaedleri* or *Akkermansia muciniphila* [8,46]. Mucin glycosylation is an important element of mucosal barrier function in the intestine, and genetically or environmentally-induced alteration of mucin glycosylation will increase susceptibility to IBD [14].

## Conclusion

This is the first report describing *O*-glycans from mice with a disrupted intestinal barrier due to genetic ROS deficiency. Our combined methodology consisting of comprehensive *O*-glycan analysis of total mucin extract from mouse colon and ileum using HILIC-UPLC in combination with exoglycosidase enzyme digestions and MS enabled us to identify five novel glycans in ileum. Our results in *Cyba*[nmf333] mice are consistent with reported *O*-glycan changes in intestinal mucins associated with colitis, expanding the insight into IBD pathogenesis.

## Supporting information

**S1 Fig. Separation of HILIC-UPLC profiles into glycan peaks; numbers of most abundant peaks are shown.**
(TIF)

**S1 Table. Reproducibility of microwave assisted nonreductive β-elimination in conjunction with HILIC-UPLC-FLR.** Coefficient of Variation calculation for 123 integrated peaks in pooled mucins from tissues.
(XLSX)

**S2 Table.** A) Structural characterization of the *O*-linked glycans from colon of WT and *Cyba* mut mice. B) Structural characterization of the *O*-linked glycans from ileum of WT and *Cyba* mut mice. The novel glycans are highlighted. Summary of common features consisting of pooled Individual glycans.
(XLSX)

**S3 Table.** A) HILIC-UPLC and exoglycosidase digestion of colon pool. B) HILIC-UPLC and exoglycosidase digestion of ileum pool.
(XLSX)

**S4 Table. List of individual glycan peaks (GPs) in each colon and ileum pool.** The novel glycans including their digests are highlighted.
(XLSX)

## Acknowledgments

The authors would like to thank the late Dr Mary Gallagher, who supervised the mucin purification from mouse mucus samples; Suisheng Zhang and Maurice O'Mara for mouse colony maintenance and genotyping. Structural characterization of reduced *O*-glycans was done in BIOMS, the Swedish infrastructure for mass spectrometry, supported by the Swedish Research Council.

## Author Contributions

**Conceptualization:** Radka Saldova, Ulla G. Knaus.

**Data curation:** Radka Saldova, Kristina A. Thomsson, Hayden Wilkinson, Maitrayee Chatterjee, Ashish K. Singh, Niclas G. Karlsson, Ulla G. Knaus.

**Formal analysis:** Radka Saldova, Kristina A. Thomsson, Hayden Wilkinson, Maitrayee Chatterjee, Ashish K. Singh, Niclas G. Karlsson, Ulla G. Knaus.

**Funding acquisition:** Ulla G. Knaus.

**Investigation:** Radka Saldova, Kristina A. Thomsson, Hayden Wilkinson, Maitrayee Chatterjee, Ashish K. Singh, Niclas G. Karlsson, Ulla G. Knaus.

**Methodology:** Radka Saldova, Kristina A. Thomsson, Hayden Wilkinson, Maitrayee Chatterjee, Niclas G. Karlsson.

**Project administration:** Radka Saldova, Ulla G. Knaus.

**Resources:** Ulla G. Knaus.

**Supervision:** Radka Saldova, Niclas G. Karlsson, Ulla G. Knaus.

**Visualization:** Radka Saldova, Kristina A. Thomsson, Ashish K. Singh, Ulla G. Knaus.

**Writing – original draft:** Radka Saldova, Ulla G. Knaus.

**Writing – review & editing:** Radka Saldova, Kristina A. Thomsson, Hayden Wilkinson, Maitrayee Chatterjee, Ashish K. Singh, Niclas G. Karlsson, Ulla G. Knaus.

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
