## [Decision Letter · Decision Letter 0]

29 Nov 2023

PONE-D-23-28597Caratterizzazione del glicoma O intestinale nella carenza di specie reattive dell'ossigenoPLOS ONE 

Dear Dr. Fahey (Saldova),

Thank you for submitting your manuscript to PLOS ONE. After careful consideration, we feel that it has merit but does not fully meet PLOS ONE’s publication criteria as it currently stands. Therefore, we invite you to submit a revised version of the manuscript that addresses the points raised during the review process.

We look forward to receiving your revised manuscript.

Kind regards,

Donatella Mentino

Academic Editor

PLOS ONE

“This work was supported by Science Foundation Ireland (SFI, https://www.sfi.ie/) co-funded under the European Regional Development Fund under grant number 13/RC/2073 (RS, HW), Science Foundation Ireland 16/IA/4501 (UGK).”

Additional Editor Comments:

The reviewers recommend

reconsideration of your manuscript following minor revision and

modification. I invite you to resubmit your manuscript after addressing the

comments below.

I Have some questions:

1)Why did you sacrifice after 14-16 weeks?

2)Indicate arrows on the figures.

3)Has the UEA been confirmed on how many slides? Have gender differences been highlighted?

Reviewers' comments:

Reviewer's Responses to Questions

**Comments to the Author**

1. Is the manuscript technically sound, and do the data support the conclusions?

Reviewer #1: Yes

Reviewer #2: Yes

2. Has the statistical analysis been performed appropriately and rigorously? 

Reviewer #1: Yes

Reviewer #2: Yes

3. Have the authors made all data underlying the findings in their manuscript fully available?

Reviewer #1: Yes

Reviewer #2: Yes

4. Is the manuscript presented in an intelligible fashion and written in standard English?

Reviewer #1: Yes

Reviewer #2: Yes

5. Review Comments to the Author

Reviewer #1: The study presents a characterization of glycome in ROS-deficient mutant mice by multiple approaches. Goals are correctly addressed, and material and methods are exhaustive. Results are clearly reported and expressed and conclusions are sound. Thus, I reccomend to accept the paper with minor revisions, as follows:

1) Line 202, correct "Ffollowing" and insert a period before.

2) Fig. 1 would be clearer if some arrows were inserted to indicate goblet cells and MUC2 layers.

3) Fig. 3 A,C are too small and characters cannot be read, they should be larger

Reviewer #2: The entire introduction is well-founded with adequate references. The objective of the study is clear and well describe.

Authors will need to review typos and punctuation

results:

Have you noticed any differences in mucin expression between genders?

6. PLOS authors have the option to publish the peer review history of their article (what does this mean?). If published, this will include your full peer review and any attached files.

Reviewer #1: No

Reviewer #2: No

---

## [Author Response · Author response to Decision Letter 0]

13 Dec 2023

Dear Editors, 

We thank the reviewers for their helpful and constructive suggestions on our paper entitled:- “Characterization of intestinal O-glycome in reactive oxygen species deficiency”. 

We hope that both you and the reviewers find this revised manuscript improved and now acceptable for consideration for publication. 

The revised draft of the manuscript along with the comments in this letter addresses the concerns raised by the reviewers. Our specific point-by-point responses are found below and we have numbered the reviewers' queries to assist in the review process.

Additional requirements were addressed, namely-

Style was updated according to the instructions, typos were corrected.

Remaining data were deposited to Glycopost project ID no. GPST000386 (https://glycopost.glycosmos.org/entry/GPST000386).

Response to Editor and Reviewers

Response to Editor

1)Why did you sacrifice after 14-16 weeks?

We sacrificed older mice as recommended for mucus analysis as mouse mucus composition has stabilized after 12 weeks of age. 

2)Indicate arrows on the figures

As requested, we have added arrows for goblet cells and the MUC2 layer and indicated this in the Figure legend Fig 1.

3)Has the UEA been confirmed on how many slides? Have gender differences been highlighted?

UEA staining was performed on colonic tissues of 4 mice/group (both genders, 3-4 slides with 4 tissues per slide/mouse) and representative images are shown. The Cybanmf333 mice were thoroughly characterized in Mucosal Immunol. 2019, 12(6):1316-1326 using female and male mice, and no apparent gender-specific differences were observed in mucus (Fig.4, Figs S6, S7). Therefore, as indicated, mucus was harvested from both genders and combined for analysis.

Response to Reviewer 1

1)Line 202, correct "Ffollowing" and insert a period before. 

The text has been corrected throughout.

2)Fig 1 would be clearer if some arrows were inserted to indicate goblet cells and MUC2 layers.

As requested, we have added arrows for goblet cells and the MUC2 layer and indicated this in the Figure legend Fig 1.

3)Fig. 3 A,C are too small and characters cannot be read, they should be larger

The small characters on Figure 3A and C were increased in size.

Response to Reviewer 2

1)Authors will need to review typos and punctation

The text has been corrected throughout.

2)Have you noticed any differences in mucin expression between genders?

The Cybanmf333 mice were thoroughly characterized in Mucosal Immunol. 2019, 12(6):1316-1326 using female and male mice, and no apparent gender-specific differences were observed in mucus (Fig.4, Figs S6, S7). Therefore, as indicated, mucus was harvested from both genders and combined for analysis.

Kind Regards,

Radka Saldova et al.

---

## [Editor Report · Decision Letter 1]

3 Jan 2024

Caratterizzazione del glicoma O intestinale nella carenza di specie reattive dell'ossigeno 

PONE-D-23-28597R1

Caro dottore. Radka Fahey (Saldova)

Siamo lieti di informarti che il tuo manoscritto è stato giudicato scientificamente idoneo per la pubblicazione e sarà formalmente accettato per la pubblicazione una volta che soddisferà tutti i requisiti tecnici in sospeso.

Entro una settimana riceverai un'e-mail con i dettagli delle modifiche richieste. Una volta risolti questi problemi, riceverai una lettera formale di accettazione e il tuo manoscritto verrà programmato per la pubblicazione.

Poco dopo l'accettazione formale seguirà la fattura per il pagamento. Per garantire un processo efficiente, accedi a Gestione editoriale all'indirizzo http://www.editorialmanager.com/pone/, fai clic sul collegamento "Aggiorna le mie informazioni" nella parte superiore della pagina e ricontrolla che le informazioni dell'utente siano aggiornate. ad oggi. Se hai domande relative alla fatturazione, contatta direttamente il nostro dipartimento Fatturazione autori all'indirizzoauthorbilling@plos.org.

Se la tua istituzione o le tue istituzioni dispongono di un ufficio stampa, informali del tuo prossimo articolo per aiutarli a massimizzarne l'impatto. Se stanno preparando materiale per la stampa, informa il nostro team stampa il prima possibile, entro e non oltre 48 ore dalla ricezione dell'accettazione formale. Il tuo manoscritto rimarrà sotto severo embargo stampa fino alle 14:00, ora di New York, della data di pubblicazione. Per ulteriori informazioni, contattare onepress@plos.org.

Cordiali saluti,

Donatella Mentino 

Redattore Accademico 

PLOS ONE

Commenti aggiuntivi dell'editor (facoltativi):

Commenti dei revisori:

---

## [Editor Report · Acceptance letter]

14 Jan 2024

PONE-D-23-28597R1 

PLOS ONE

Dear Dr. Saldova, 

I'm pleased to inform you that your manuscript has been deemed suitable for publication in PLOS ONE. Congratulations! Your manuscript is now being handed over to our production team.

Kind regards, 

on behalf of

Dr. Donatella Mentino 

Academic Editor

PLOS ONE